# Translation, Cultural Adaptation, and Validation into Romanian of the Myeloproliferative Neoplasm Symptom Assessment Form—Total Symptom Score (MPN-SAF TSS or MPN-10) Questionnaire

**DOI:** 10.3390/jcm13113284

**Published:** 2024-06-02

**Authors:** Mihnea-Alexandru Găman, Robyn Marie Scherber, Iulia Ursuleac, Ana Manuela Crişan, Sorina Nicoleta Bădeliţă, Bogdan Octavian Ionescu, Alexandra Elena Ghiaur, Melen Brînză, Nicoleta Pîrciulescu, Toma Octavian Lascăr, Camelia Cristina Diaconu, Amelia Maria Găman, Daniel Coriu

**Affiliations:** 1Faculty of Medicine, “Carol Davila” University of Medicine and Pharmacy, 050474 Bucharest, Romania; mihnea-alexandru.gaman@drd.umfcd.ro (M.-A.G.); iuliaursuleac@yahoo.com (I.U.); crisananamanuela@yahoo.com (A.M.C.); or drcameliadiaconu@gmail.com (C.C.D.); daniel_coriu@yahoo.com (D.C.); 2Department of Hematology, Centre of Hematology and Bone Marrow Transplantation, Fundeni Clinical Institute, 022328 Bucharest, Romania; sorinabadelita@gmail.com (S.N.B.); ionescu.bogdan44@yahoo.com (B.O.I.); ghiaur.alexandra@gmail.com (A.E.G.); melen.brinza@yahoo.com (M.B.); nicoletapirciulescu@yahoo.com (N.P.); tomalascar@yahoo.com (T.O.L.); 3Department of Cellular and Molecular Pathology, Stefan S. Nicolau Institute of Virology, Romanian Academy, 030304 Bucharest, Romania; 4Department of Hematology/Oncology, UT Health San Antonio, MD Anderson Cancer Center, San Antonio, TX 78229, USA; scherber@uthscsa.edu; 5Internal Medicine Clinic, Clinical Emergency Hospital of Bucharest, 105402 Bucharest, Romania; 6Department of Pathophysiology, University of Medicine and Pharmacy of Craiova, 200349 Craiova, Romania; 7Clinic of Hematology, Filantropia City Hospital, 200143 Craiova, Romania

**Keywords:** quality of life, myeloproliferative neoplasms, polycythemia vera, essential thrombocythemia, myelofibrosis, patient-reported outcome measures, symptom burden, MPN Symptom Assessment Form, MPN-10, MPN-SAF

## Abstract

**Background:** Patients with myeloproliferative neoplasms (MPNs) experience a high disease-related symptom burden. A specific instrument to evaluate quality of life (QoL), i.e., the MPN Symptom Assessment Form Total Symptom Score (MPN-SAF TSS; MPN-10), was developed. We conducted the translation, cultural adaptation, and validation into Romanian of the MPN-10. **Methods:** We translated the MPN-10 and tested its psychometric properties. **Results:** We recruited 180 MPN patients: 66 polycythemia vera (36.67%), 61 essential thrombocythemia (33.89%), 51 primary and secondary myelofibrosis (SMF) (28.33%), and 2 MPN-unclassifiable (1.11%). The mean TSS was 19.51 ± 16.51 points. Fatigue, inactivity, and concentration problems were the most cumbersome symptoms. We detected scoring differences between MPN subtypes regarding weight loss (*p* < 0.001), fatigue (*p* = 0.006), early satiety (*p* = 0.007), night sweats (*p* = 0.047), pruritus (*p* = 0.05), and TSS (*p* = 0.021). There were strong positive associations between TSS and inactivity, fatigue, and concentration problems, and moderate negative correlations between QoL scores and all MPN-10 items. Cronbach’s α internal consistency coefficient was 0.855. The Kaiser–Meyer–Olkin construct validity test result was 0.870 and the Bartlett Sphericity Test was significant (*p* < 0.001). Symptom scores were loaded into one single factor according to the exploratory factor analysis. **Conclusions:** The Romanian MPN-10 version displayed excellent psychometric properties and is a reliable instrument for assessing symptom burden and QoL in Romanian MPN patients.

## 1. Introduction

Polycythemia vera (PV), essential thrombocythemia (ET), and myelofibrosis (MF) are clonal disorders of the hematopoietic stem cells included by the World Health Organization (WHO) and the International Consensus Classification (ICC) of myeloid neoplasms and acute leukemias in the category of classical myeloproliferative neoplasms (MPNs) [1]. MPNs are defined by an uncontrolled proliferation of terminally differentiated cells of the myeloid lineage, the presence of driver mutations in the Janus kinase 2 (JAK2), calreticulin (CALR) or thrombopoietin receptor (MPL or TPOR) genes, an absence of the BCR::ABL1 disease marker, and an elevated risk of thrombotic as well as bleeding events, in addition to a propensity for the progression of PV or ET to secondary MF (SMF) or of all classical MPNs to evolve into acute myeloid leukemia (AML) [1,2,3,4].

Moreover, patients diagnosed with MPNs experience a myriad of symptoms that contribute to the burden of disease and result in an overall reduced health-related quality of life (QoL). MPN subjects are often stressed about the presence of constitutional, vascular, or disease-related symptoms, as well as splenomegaly-linked complaints [5]. Therefore, a psychometric instrument was developed by a team of international experts and later published in 2012 by Dr. Robyn (Emmanuel) Scherber and collaborators to accurately evaluate QoL and disease burden in BCR::ABL1-negative MPNs. The aforementioned instrument, termed the Myeloproliferative Neoplasm Symptom Assessment Form Total Symptom Score (MPN-SAF TSS) or simply MPN-10, is an abbreviated symptom burden scoring system particularly developed to assess QoL in individuals living with classical BCR::ABL1-negative MPNs, i.e., PV, ET, primary (PMF) or secondary myelofibrosis (SMF), and MPN-unclassifiable (MPNu) [6].

The MPN-10 has been translated, culturally adapted, and validated to various languages, e.g., French, German, Italian, Spanish, Portuguese, Dutch, Swedish, etc., and has become a useful tool in evaluating patient-reported outcomes (PROs), including in the setting of randomized clinical trials (RCTs) [5,6,7,8,9]. However, a Romanian version of the MPN-10 has not been yet made available, even though there are no other psychometric instruments available in Romanian to measure symptom burden and QoL in patients diagnosed with MPNs.

Herein, we aimed to provide the translation, cultural adaptation, and validation of the Romanian version of the MPN-10 (RO-MPN-10) and present, to our knowledge, a first report on the QoL status of MPN patients from Romania.

## 2. Materials and Methods

**Participants, setting, and study details**. The investigation was conducted in the Department of Hematology, Center of Hematology and Bone Marrow Transplantation, Fundeni Clinical Institute, Bucharest, Romania, and in the Internal Medicine Clinic, Clinical Emergency Hospital of Bucharest, Bucharest, Romania. The study was approved by the ethics committee/ethics council of both institutions (reference number 40542/07.06.2021, approved on 27 May 2021; and reference number 3982/13.04.2021, approved on 13.04.2021, respectively) and was carried out between June 2021 and December 2023. Patients were included in the study group if 1. they were diagnosed with BCR::ABL1-negative MPNs based on the World Health Organization diagnostic criteria; 2. they were aged ≥18 years; 3. they did not suffer from psychiatric disorders or cognitive impairment; 4. they were literate and able to self-administer the questionnaire; and 5. they signed the informed consent form. Patients with other solid or hematological malignancies, acute infections, or decompensated comorbidities (e.g., heart failure, chronic obstructive pulmonary disorder, etc.) which could have altered the grading of MPN-related symptoms were excluded from the investigation and therefore were not invited to complete the RO-MPN-10 questionnaire.

**Instrument**. The MPN-SAF TSS consists of one item (worst fatigue during the last 24 h) derived from the Brief Fatigue Inventory [10] and nine items (early satiety, abdominal discomfort, inactivity, concentration problems, night sweats, itching/pruritus, bone pain, unintentional weight loss in the last 6 months and fever >37.7 Celsius degrees) designed based on the frequency of symptoms reported by individuals living with MPNs. Patients rate each item on a scale from 0 points (i.e., the symptom is absent) to 10 points (i.e., the severity of the symptom is the worst imaginable), and the final total symptom score is calculated as the sum of the ten individual symptom scores. MPN subjects were viewed as symptomatic if their scores were higher than 0 points. Symptom severity was graded as moderate if the individual symptom score ranged from 4 points to 6 points and severe if the individual symptom score was equal to or higher than 7 points. The TSS was calculated as the average of the observed individual symptom scores multiplied by 10, and thus the TSS ranged from 0 points to 100 points. In our assessment, MPN patients were regarded as “clinically deficient” if they experienced a decline in their overall QoL of at least 4 points (QoL score ≤ 6 points). Subjects diagnosed with MPNs were invited to self-administer the RO-MPN-10 during their visit to the outpatient clinic or during their hospitalization in the clinical ward. 

**Translation, validation, and cultural adaptation into Romanian**. The first author (M.-A.G.) was authorized by the original authors of the MPN-SAF TSS to develop the Romanian version of the questionnaire and one of the original authors of the instrument was also involved in the project (R.M.S.). The translation, validation, and cultural adaptation of the instrument were developed in agreement with the international guidelines. Briefly, the questionnaire was translated into Romanian by two native speakers of Romanian and then back-translated into English by two researchers with native proficiency in English who did not have access to the original version of the instrument. The final version of the RO-MPN-10 was achieved by consensus and distributed to a small number of individuals diagnosed with MPNs (n = 10) to test semantic equivalence, and the final version agreed upon was distributed to Romanian MPN patients. The minimum study sample was calculated to be 80–100 subjects based on the Patient-Reported Outcomes (PRO) guidelines based on the fact that at least 8–10 responses were needed for each item of the instrument item (8–10 responses/item × 10 items in the MPN-10 = 80–100 participants) [11].

**Statistical analysis**. Statistical analysis was performed using JASP version 0.18.3.0 (Eric-Jan Wagenmakers (room G 0.29), Department of Psychological Methods, University of Amsterdam, Nieuwe Achtergracht 129B, Amsterdam, The Netherlands). Descriptive statistics were used to report data regarding age, gender, genetics, and MPN subtype distribution, as well as incidence and severity of symptoms and QoL scores. Inter-group means and differences in incidences were compared using ANOVA and chi-squared tests, respectively. Cronbach’s alpha coefficient was used to assess the internal consistency of RO-MPN-10 and values higher than 0.7 were considered as good in line with previous reports [7]. Convergent validity was evaluated using Pearson/Spearman correlation coefficients, with correlation coefficients higher than or equal to 0.70 indicating strong associations and values higher than or equal to 0.40 but lower than 0.70 indicating moderate associations [7]. Exploratory factor analysis was performed to assess construct validity and factors with eigenvalues higher than 1 were retained. The appropriateness of the data for the aforementioned analysis was tested using the Kaiser–Meyer–Olkin (KMO > 0.70 is recommended) and the Bartlett Sphericity Tests (*p* < 0.05), respectively [7]. The threshold for statistical significance was set at 0.05 (*p* < 0.05). The Kolmogorov–Smirnov test was used to check whether the study sample was normally distributed.

## 3. Results

The study group consisted of 180 patients diagnosed with MPNs, namely PV (36.67%; n = 66), ET (33.89%; n = 61), MF (28.33%; n = 51), and MPN-unclassifiable (MPNu) (1.11%; n = 2). Individuals with MF (n = 51) suffered either from PMF (n = 34) or SMF, specifically post-PV MF (n = 7) or post-ET MF (n = 10). The mean age ± standard deviation (SD) of the study participants was 62.75 ± 12.36 years, and 54.44% of them were female. JAK2V617F was detected in 141 MPN subjects (78.33%), whereas 10 subjects displayed CALR type I/II mutations (5.55%). 

The overall TSS for the entire MPN cohort was 19.51 ± 16.51 points. Participants diagnosed with MF experienced elevated TSS values (24.84 ± 19.63 points) versus those with PV (18.95 ± 14.38 points) and ET (15.30 ± 14.88 points). TSS was higher in individuals with SMF, i.e., post-PV (29.86 ± 15.39 points) and post-ET (24.50 ± 27.39 points) MF as compared to PMF (23.91 ± 18.16 points), respectively. Of note, patients diagnosed with MPNu registered the highest TSS (30.50 ± 3.54 points); however, the MPNu subgroup only consisted of two subjects.

Overall, in the entire MPN cohort, fatigue (3.67 ± 2.77 points), inactivity (2.73 ± 2.71 points), and concentration problems (2.39 ± 2.55 points) were the symptoms scoring the highest on the TSS scale. Subjects suffering from PV rated fatigue (3.73 ± 2.64 points), inactivity (2.74 ± 2.49 points), and night sweats (2.69 ± 2.92 points) as the symptoms affecting them the most. Similarly, ET patients rated fatigue (2.80 ± 2.52 points) as the highest symptom on the TSS scale, followed by bone pain (2.16 ± 2.88 points), inactivity (2.16 ± 2.35 points), and concentration problems (2.16 ± 2.36 points). Individuals living with MF complained the most about the presence of fatigue (4.51 ± 2.96 points), inactivity (3.18 ± 3.12 points), and bone pain (2.86 ± 3.40 points). PMF subjects complained the most about fatigue (4.18 ± 2.81 points), early satiety (2.79 ± 2.73 points), and night sweats (2.85 ± 3.22 points). Patients with post-ET MF rated fatigue (5.10 ± 3.75 points), inactivity (3.90 ± 4.36 points), and bone pain (3.10 ± 4.12 points) highest on the symptom scale, whereas subjects suffering from post-PV MF reported that fatigue (5.29 ± 2.50 points), bone pain (4.29 ± 2.93 points), and inactivity (4.14 ± 2.73 points) had a more meaningful impact on their QoL. Table 1 depicts the MPN-SAF individual symptom scores and incidences, overall QoL, and TSS for the individuals diagnosed with PV, ET, and MF enrolled in our assessment. Table 2 summarizes the MPN-SAF individual symptom scores and incidences, overall QoL, and TSS in subjects diagnosed with MF (PMF vs. SMF) who partook in our investigation.

In total, 39 MPN patients (21.66%) were clinically deficient, producing QoL scores ≤ 6 points: 15 subjects suffered from PV, 12 from MF (8 PMF; 4 SMF, of whom 2 had post-PV and 2 had post-ET MF), and 12 from ET. 

We detected notable differences between MPN subtypes in terms of individual symptom scores for unintentional weight loss (*p* < 0.001; PMF vs. PV, *p* = 0.001; PMF vs. TE, *p* = 0.002), worst fatigue (*p* = 0.006; SMF vs. ET, *p* = 0.009), early satiety (*p* = 0.007; PMF vs. PV, *p* = 0.011), night sweats (*p* = 0.047), and pruritus (*p* = 0.05), as well as for the TSS (*p* = 0.021). 

In addition, our study demonstrated positive strong associations between TSS and inactivity (r = 0.741, *p* < 0.001), worst fatigue (r = 0.734, *p* < 0.001), and concentration problem (r = 0.709; *p* < 0.001) scores, respectively. Moreover, we discovered moderate negative correlations between QoL and all MPN-10 items (*p* < 0.001). Figure 1 depicts the correlations detected between the items of the MPN-SAF TSS.

The internal consistency of the Romanian version of the questionnaire was excellent (Cronbach’s α coefficient = 0.855), ranging from 0.830 for pruritus to 0.861 for fever. Construct validity (Kaiser-Meyer-Olkin test = 0.870, Bartlett Sphericity Tests *p* < 0.001) was solid. Exploratory factor analysis demonstrated that the symptoms loaded into one single factor. Test–retest data were available for 63 MPN patients, revealing that TSS scores did not change significantly between visits (*p* = 0.24), thus demonstrating that RO-MPN-10 is a reliable instrument to evaluate QoL at different time points. However, the test–retest method highlighted an increase in TSS and a decrease in QoL scores, respectively, in patients with MPNs who progressed to SMF or AML, indicating that RO-MPN-10 can discriminate between MPN subtypes and capture changes in disease biology.

## 4. Discussion

Herein, we provided the translation, validation, and cultural adaptation of the MPN-SAF TSS questionnaire into Romanian. Based on our results, the RO-MPN-10 seems a reliable instrument to assess the burden of symptoms and the QoL of Romanian patients diagnosed with BCR::ABL1-negative MPNs. Existing instruments designed to evaluate QoL in subjects living with malignant disorders mostly apply to individuals diagnosed with solid cancers, with the number of disease-specific questionnaires that can be applied to patients with hematological disorders remaining limited. Thus, making use of a validated, disease-specific instrument such as the MPN-10 has become extremely relevant in the modern management of MPNs, particularly as collecting data on PROs has been demonstrated to influence current practices in and out of the setting of RCTs by guiding shared decision making and patient–physician communication [12,13,14].

As expected, our assessment highlights that patients diagnosed with MF, and in particular SMF, displayed the highest TSS and the most notable QoL reduction amongst MPNs, in agreement with the results of the international validation of the MPN-10 and language-specific validation studies [6,7]. Significant differences were noted between MPN subtypes in terms of the overall burden of symptoms, as well as the presence of pruritus, night sweats, and other complaints. In particular, subjects suffering from MF graded unintentional weight loss, fatigue, and early satiety higher on the symptom scale versus individuals who were living with other MPN subtypes, in concordance with previous reports [14,15]. Our MPN cohort graded fatigue, inactivity, and concentration problems as the most cumbersome symptoms. Fatigue was indeed the most meaningful symptom reported, irrespective of MPN subtype, scoring the highest for PV, ET, PMF, and SMF, closely followed by inactivity (rated second by PV, ET, and post-ET MF and third by post-PV MF patients, respectively) and bone pain (rated second by ET and post-PV MF and third by post-ET MF). Night sweats (rated third by PV and PMF), concentration problems (rated third by ET), and early satiety (rated second by PMF) also ranked high in the symptom scoring system. Our findings are relatively similar to those presented by previous publications. Guarana et al. revealed that Brazilian subjects with MPNs ranked fatigue, inactivity, and concentration problems as the most severe disease symptoms. In their assessment, fatigue, early satiety, and concentration problems/abdominal discomfort were the most cumbersome symptoms in MF, whereas PV patients complained the most about pruritus, inactivity, and bone pain, and ET subjects about fatigue, concentration problems, and inactivity [7]. Interestingly, in our investigation, individuals suffering from PV did not rank itching as a symptom severely impacting their QoL, possibly because it was well controlled by the use of risk-adapted therapy. However, our MPN cohort frequently complained of non-arthritic bone pain, raising the question of whether the recruited subjects were able to discriminate accurately between bone pain related to arthritis, aging, or caused by other degenerative conditions and disease-related bone pain. In the prospective international assessment of the MPN-10, data collected from nearly 1500 MPN subjects stressed fatigue, early satiety, and concentration problems as the most severe symptoms in the entire cohort. MF patients ranked fatigue, early satiety, and inactivity as the most severe symptoms, whereas PV subjects ranked fatigue, itching, and concentration problems, and individuals with ET ranked fatigue, early satiety, and concentration problems as the most cumbersome features of their disease [6]. Thus, our findings revolve, with some differences, around the same spectrum of symptom severity and incidence as previous investigations.

Moreover, we detected strong positive associations between TSS and several of the symptoms graded as severe by our patients, namely inactivity, fatigue, and concentration problems, as well as moderate negative correlations between QoL scores and all MPN-10 items. In addition, with small exceptions, there were moderate or minor statistically significant associations between the items of the RO-MPN-10. For example, fatigue scores did not correlate with fever scores; however, they were weakly associated with itching, pruritus, and bone pain, strongly associated with inactivity, and moderately associated with the remaining items of the RO-MPN-10. Similarly, correlations between RO-MPN-10 items remained moderate or to a lesser extent weak, except for the fatigue–fever, fever–early satiety, fever–inactivity, and itching–weight loss pairs, for which we did not demonstrate any associations, probably because fever was a symptom reported by approximately 6% of the Romanian MPN cohort. Similarly, convergent validity analyses conducted by other scientists have highlighted moderate correlations between fatigue and inactivity (r = 0.51, *p* < 0.001 versus r = 0.609, *p* < 0.001 in our study), early satiety (r = 0.51, *p* < 0.001 versus r = 0.407, *p* < 0.001 in our study), concentration problems (r = 0.40, *p* < 0.001 versus r = 0.515, *p* < 0.001 in our study), or QoL (r = 0.45, *p* < 0.001 versus r = −0.528, *p* < 0.001 in our study), as well as between abdominal discomfort and early satiety (r = 0.59, *p* < 0.001 versus r = 0.676, *p* < 0.001 in our study), abdominal discomfort and fatigue (r = 0.53, *p* < 0.001 versus r = 0.466, *p* < 0.001 in our study), abdominal discomfort and inactivity (r = 0.49, *p* < 0.001 versus r = 0.465, *p* < 0.001 in our study), and weight loss and fever (r = 0.42, *p* < 0.001 versus r = 0.242, *p* < 0.001 in our study) [7]. Of note, RO-MPN-10 items correlated negatively with QoL scores because we performed a change in the method of evaluating QoL, as suggested by both patients and physicians in the testing phase of the RO-MPN-10. Specifically, in several other versions of the questionnaire, patients attributed 10 points to the QoL item if their QoL was the worst (“as bad as it can be”), whereas 0 points were awarded to the item if their QoL was the best (“as good as it can be”). However, since the MPN-10 is not amongst the 10 items included in the validation of the MPN-10, this change did not influence the translation or validation procedure and was viewed as an integrated part of the cultural adaptation of the MPN-10.

The internal consistency of the Romanian version of the questionnaire was excellent (Cronbach’s α coefficient = 0.855), ranging from 0.830 for pruritus to 0.861 for fever. The construct validity (Kaiser–Meyer–Olkin test = 0.870, Bartlett Sphericity Tests *p* < 0.001) was solid. Exploratory factor analysis demonstrated that the symptoms loaded into one single factor.

Our study has some strengths and limitations. The main advantages of our research include the fact that this is the first report regarding the QoL status of MPN patients from Romania and the psychometric properties of the Romanian version of the MPN-10 were excellent, proving that the translated version of the survey is an adequate tool to assess symptom burden in subjects from this geographical area living with these hematological malignancies. Limitations include the fact that the study only included one hematology reference center and that retesting was only performed in approximately a third of the recruited cohort.

## 5. Conclusions

Our study reveals that the Romanian version of the MPN-SAF TSS questionnaire displayed excellent psychometric properties and can be a reliable instrument for the assessment of symptom burden and QoL in MPN patients.

## Figures and Tables

**Figure 1 jcm-13-03284-f001:**
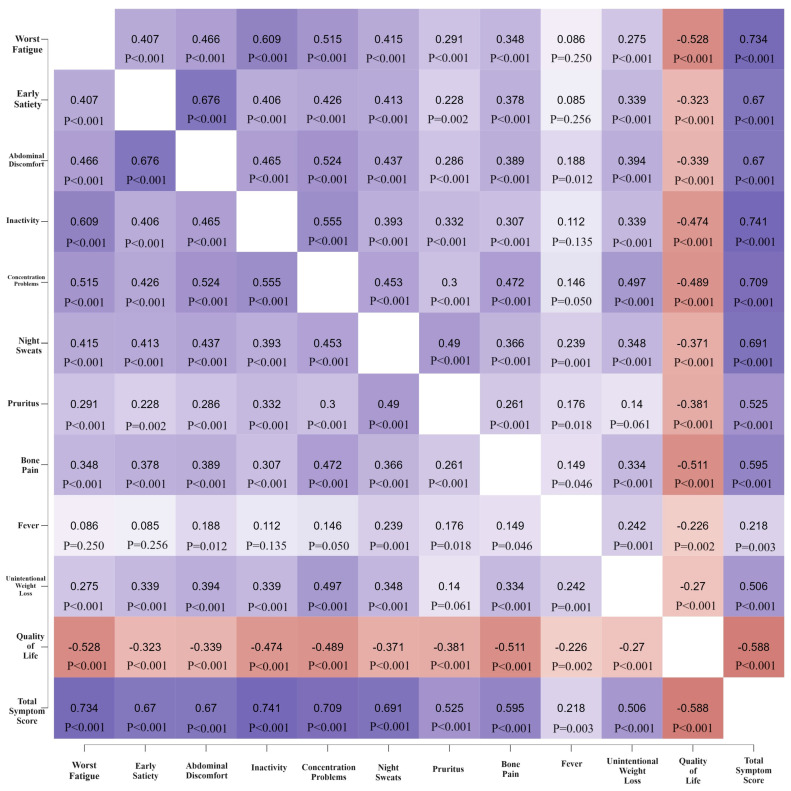
Correlation heatmap between MPN-10 items, QoL, and TSS. Legend: QoL, quality of life. TSS, total symptom score.

**Table 1 jcm-13-03284-t001:** Myeloproliferative neoplasm symptom assessment scores and symptom incidences in patients with polycythemia vera, essential thrombocythemia, and myelofibrosis.

Symptom	PV (n = 66)			ET (n = 61)			MF (n = 51)			MPNs Overall (n = 180)			*p*-Values
	Mean	SD	Incidence	Mean	SD	Incidence	Mean	SD	Incidence	Mean	SD	Incidence	
Worst fatigue	3.73	2.64	92.42%	2.80	2.52	86.89%	4.51	2.96	86.27%	3.67	2.77	88.89%	*p* = 0.004; MF vs. ET, *p* = 0.002
Early satiety	1.23	1.80	48.49%	1.61	2.42	52.46%	2.75	2.84	70.59%	1.84	2.45	56.67%	*p* = 0.002; MF vs. PV, *p* = 0.001; MF vs. ET, *p* = 0.02
Abdominal discomfort	1.20	1.90	48.49%	1.23	2.12	40.98%	2.12	2.81	58.82%	1.49	2.32	48.89%	*p* = 0.057
Inactivity	2.74	2.49	78.79%	2.16	2.35	68.85%	3.18	3.12	72.55%	2.73	2.71	73.89%	*p* = 0.128
Problems with concentration	2.52	2.43	71.21%	2.16	2.36	63.93%	2.55	2.93	66.67%	2.39	2.55	67.22%	*p* = 0.663
Night Sweats	2.68	2.92	72.73%	1.43	2.65	44.26%	2.82	3.24	64.71%	2.27	2.97	60.56%	*p* = 0.018; MF vs. ET, *p* = 0.029
Itching (pruritus)	2.11	2.55	66.67%	0.93	2.23	22.95%	1.49	2.37	43.14%	1.51	2.42	44.44%	*p* = 0.024; PV vs. ET, *p* = 0.023
Bone pain	1.77	2.49	60.61%	2.16	2.88	57.38%	2.86	3.40	66.67%	2.19	2.92	60.56%	*p* = 0.133
Fever (>37.7 Celsius)	0.28	1.44	6.06%	0.05	0.28	3.28%	0.35	1.28	9.80%	0.22	1.12	6.11%	*p* = 0.321
Unintentional weight loss last 6 months	0.72	1.32	34.85%	0.75	1.70	31.15%	2.27	3.23	52.94%	1.22	2.28	39.44%	*p* < 0.001; MF vs. PV, *p* < 0.001; MF vs. ET, *p* < 0.001
Overall QoL	7.39	1.39		7.61	1.79		7.25	2.27		7.44	1.81		*p* = 0.584
TSS	18.95	14.38		15.30	14.88		24.84	19.63		19.51	16.51		*p* = 0.008; MF vs. ET, *p* = 0.004

Legend: QoL, quality of life. TSS, total symptom score. PV, polycythemia vera. ET, essential thrombocythemia. MF, myelofibrosis. MPNs, myeloproliferative neoplasms. n, number. SD, standard deviation.

**Table 2 jcm-13-03284-t002:** Myeloproliferative neoplasm symptom assessment scores and symptom incidences in patients with primary myelofibrosis, post-essential thrombocythemia, and post-polycythemia vera myelofibrosis.

Symptom	PMF (n = 34)			Post-ET MF (n = 10)			Post-PV MF (n = 7)			MF Overall (n = 51)			*p*-Values
	Mean	SD	Incidence	Mean	SD	Incidence	Mean	SD	Incidence	Mean	SD	Incidence	
Worst fatigue	4.18	2.81	85.29%	5.10	3.75	80.00%	5.29	2.50	100.00%	4.51	2.96	86.27%	*p* = 0.527
Early satiety	2.79	2.73	76.47%	2.00	3.09	40.00%	3.57	3.21	85.71%	2.75	2.84	70.59%	*p* = 0.534
Abdominal discomfort	2.06	2.78	58.82%	2.30	3.77	40.00%	2.14	1.35	85.71%	2.12	2.81	58.82%	*p* = 0.972
Inactivity	2.76	2.77	70.59%	3.90	4.36	60.00%	4.14	2.73	100.00%	3.18	3.12	72.55%	*p* = 0.415
Problems with concentration	2.44	2.85	64.71%	2.40	3.37	60.00%	3.29	3.04	85.71%	2.55	2.93	66.67%	*p* = 0.780
Night Sweats	2.85	3.22	61.76%	2.50	3.57	50.00%	3.14	3.34	100.00%	2.82	3.24	64.71%	*p* = 0.921
Itching (pruritus)	1.59	2.63	41.18%	1.50	1.84	50.00%	1.00	1.83	42.86%	1.49	2.37	43.14%	*p* = 0.841
Bone pain	2.50	3.28	61.76%	3.10	4.12	70.00%	4.29	2.93	85.71%	2.86	3.40	66.67%	*p* = 0.444
Fever (>37.7 Celsius)	0.35	1.32	8.82%	0.50	1.58	10.00%	0.14	0.38	14.29%	0.35	1.28	9.80%	*p* = 0.856
Unintentional weight loss last 6 months	2.47	3.40	55.88%	1.20	2.53	30.00%	2.86	3.39	71.43%	2.27	3.23	52.94%	*p* = 0.491
Overall QoL	7.47	2.08		7.00	3.13		6.57	1.90		7.25	2.27		*p* = 0.595
TSS	23.91	18.16		24.50	27.39		29.86	15.39		24.84	19.63		*p* = 0.771

Legend: QoL, quality of life. TSS, total symptom score. PV, polycythemia vera. ET, essential thrombocythemia. MF, myelofibrosis. PMF, primary myelofibrosis. n, number. SD, standard deviation.

## Data Availability

Data are available on request from the first or corresponding author.

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
