# Peer review of "Translation, Cultural Adaptation, and Validation into Romanian of the Myeloproliferative Neoplasm Symptom Assessment Form—Total Symptom Score (MPN-SAF TSS or MPN-10) Questionnaire"

_jcm, 2024, doi:10.3390/jcm13113284_

Round 1
Reviewer 1 Report
Comments and Suggestions for Authors
The Gaman et al article explores the psychometric properties and cultural adaptation to the Romanian population of a questionnaire already validated for symptomatology of patients with myeloid neoplasia. They have shown that the tool has good internal consistency and construct validity. Some suggestions to discuss with the authors would be:
- The exclusion criteria should not be the negative of the inclusion criteria.
- Lines 103-105 are repeated in the introduction.
- The calculation of the sample size is not clear.
- Was there statistical significance in the values in tables 1 and 2? The P-value for comparison between groups should be reported.
- Table 3 and fig 1 are repetitive data. Table 3 could be deleted and the correlogram with the significance inside each box.
Minor comments:
- Defines PV/TE acronym on line 62.
Author Response
Dear Editor-in-Chief
Dear Academic Editor
Dear Peer-Reviewers,
We are thankful for your valuable comments and constructive criticism regarding our paper. We have performed all requested revisions and all changes have been explained and/or highlighted in yellow in the text. We do hope that in its revised form the manuscript warrants publication in the Journal of Clinical Medicine.
Reviewer #1
The Gaman et al article explores the psychometric properties and cultural adaptation to the Romanian population of a questionnaire already validated for symptomatology of patients with myeloid neoplasia. They have shown that the tool has good internal consistency and construct validity. Some suggestions to discuss with the authors would be:
Response: Thank you for your valuable comments and constructive criticism. We do hope that in its revised form the manuscript warrants publication in the Journal of Clinical Medicine.
- The exclusion criteria should not be the negative of the inclusion criteria.
Response: Thank you for your valuable comment. We have revised the wording of the exclusion criteria as instructed.
- Lines 103-105 are repeated in the introduction.
Response: Thank you for your valuable comment. Lines 103-105 have been deleted and their content has only been retained in the Introduction.
- The calculation of the sample size is not clear.
Response: Thank you for your valuable comment. In order to validate the RO-MPN-10, we needed at least 80-100 patients as the MPN-10 has 10 items and 8-10 responses/item are required according to available guidelines on the assessment of psychometric properties of questionnaires (see reference 11).
- Was there statistical significance in the values in tables 1 and 2? The P-value for comparison between groups should be reported.
Response: Thank you for your valuable comment. P-values were reported as instructed. There was statistical significance for the results presented in Table 1. Regarding Table 2, there were no notable differences in terms of MPN-10 scores between PMF, post-PV and post-ET MF.
- Table 3 and fig 1 are repetitive data. Table 3 could be deleted and the correlogram with the significance inside each box.
Response: Thank you for your valuable comment. Table 3 was deleted. The correlation coefficients and P-values were reported in Figure 1 as instructed.
- Minor comments: - Defines PV/TE acronym on line 62.
Response: Thank you for your valuable comment. The acronyms for PV and ET have been defined in line 54. We corrected the typo, it should have been ET not TE, and rewrote “PV/TE” to read “PV or ET”.
Reviewer 2 Report
Comments and Suggestions for Authors
Dear authors, Your work is well performed. The paper quotes are for the majority of the last five years. I suggest only three minor adjustments.
· The abstract is too long, the journal suggests no more than 200 words, please reduce the length
· The result for each item of the score is reported with mean and standard deviation, but in the opinion of this reviewer, this variable should not have a Gaussian distribution and should be represented by median and range interquartile. Have you performed the Kolmogorov-Smirnov test for variables in Tables 1 and 2?
· In Table 3 and Figure 1, the abuse of abbreviation makes it difficult to read the results; please remove them as you make in Tables 1 and 2.
Author Response
Dear Editor-in-Chief
Dear Academic Editor
Dear Peer-Reviewers,
We are thankful for your valuable comments and constructive criticism regarding our paper. We have performed all requested revisions and all changes have been explained and/or highlighted in yellow in the text. We do hope that in its revised form the manuscript warrants publication in the Journal of Clinical Medicine.
Reviewer #2
Dear authors, Your work is well performed. The paper quotes are for the majority of the last five years. I suggest only three minor adjustments.
Response: Thank you for your valuable comments and constructive criticism. We do hope that in its revised form the manuscript warrants publication in the Journal of Clinical Medicine.
- The abstract is too long, the journal suggests no more than 200 words, please reduce the length
Response: Thank you for your valuable comment. The abstract has been shortened to 200 words as instructed.
- The result for each item of the score is reported with mean and standard deviation, but in the opinion of this reviewer, this variable should not have a Gaussian distribution and should be represented by median and range interquartile. Have you performed the Kolmogorov-Smirnov test for variables in Tables 1 and 2?
Response: Thank you for your valuable comment. Indeed, we have performed the Kolmogorov-Smirnov test for the variables presented in Table 1 and Table 2 and their distribution was normal.
- In Table 3 and Figure 1, the abuse of abbreviation makes it difficult to read the results; please remove them as you make in Tables 1 and 2.
Response: Response: Thank you for your valuable comment. Table 3 was deleted according to the suggestion of another peer-reviewer. Abbreviations were removed. In addition, the correlation coefficients and P-values were reported in Figure 1 as suggested by another reviewer.
Reviewer 3 Report
Comments and Suggestions for Authors
The objective of the study was to provide the translation, cultural adaptation, and validation of the Romanian version of the MPN-10, and to present first report on the QoL status of MPN patients from Romania. I find your objective very important for the assessment of symtom burden and improvement of the quality of life in this population of patients in your country.
Overall, the manuscript is well-structured and clearly written, with a good command of English and clear representation of the aim of the paper.
Abstract: The abstract summarizes the major aspects of the entire paper, including the overall purpose of the study and the research problem, the basic design of the study, major findings and a brief conclusion.
Introduction: The introduction provides a clear and concise overview of the research problem, establishing its relevance in the field.
Methods: The study methods are valid and reliable. The process of subject selection is clear. The authors employed appropriate statistical methods.
Results: Text describes in detail the obtained values and their significance in drawing conclusions. Data are presented in an appropriate way.
Discussion: Relevance and importance of the obtained results are explained well in the discussion section. The results are placed into context without being overinterpreted.
Conclusion: Conclusion answers the aims of the study and is fully supported by the results.
References: References come from reputable sources and all are cited properly throughout the manuscript.
Author Response
Dear Editor-in-Chief
Dear Academic Editor
Dear Peer-Reviewers,
We are thankful for your valuable comments and constructive criticism regarding our paper. We have performed all requested revisions and all changes have been explained and/or highlighted in yellow in the text. We do hope that in its revised form the manuscript warrants publication in the Journal of Clinical Medicine.
Reviewer #3
Comments: The objective of the study was to provide the translation, cultural adaptation, and validation of the Romanian version of the MPN-10, and to present first report on the QoL status of MPN patients from Romania. I find your objective very important for the assessment of symtom burden and improvement of the quality of life in this population of patients in your country. Overall, the manuscript is well-structured and clearly written, with a good command of English and clear representation of the aim of the paper.
Abstract: The abstract summarizes the major aspects of the entire paper, including the overall purpose of the study and the research problem, the basic design of the study, major findings and a brief conclusion.
Introduction: The introduction provides a clear and concise overview of the research problem, establishing its relevance in the field.
Methods: The study methods are valid and reliable. The process of subject selection is clear. The authors employed appropriate statistical methods.
Results: Text describes in detail the obtained values and their significance in drawing conclusions. Data are presented in an appropriate way.
Discussion: Relevance and importance of the obtained results are explained well in the discussion section. The results are placed into context without being overinterpreted.
Conclusion: Conclusion answers the aims of the study and is fully supported by the results.
References: References come from reputable sources and all are cited properly throughout the manuscript.
Response: Thank you for your valuable comments and constructive criticism. We do hope that in its revised form the manuscript warrants publication in the Journal of Clinical Medicine.